# Deciphering the Functions of Telomerase Reverse Transcriptase in Head and Neck Cancer

**DOI:** 10.3390/biomedicines11030691

**Published:** 2023-02-24

**Authors:** Tsung-Jang Yeh, Chi-Wen Luo, Jeng-Shiun Du, Chien-Tzu Huang, Min-Hung Wang, Tzer-Ming Chuang, Yuh-Ching Gau, Shih-Feng Cho, Yi-Chang Liu, Hui-Hua Hsiao, Li-Tzong Chen, Mei-Ren Pan, Hui-Ching Wang, Sin-Hua Moi

**Affiliations:** 1Division of Hematology & Oncology, Department of Internal Medicine, Kaohsiung Medical University Hospital, Kaohsiung Medical University, Kaohsiung 807, Taiwan; 2Graduate Institute of Clinical Medicine, College of Medicine, Kaohsiung Medical University, Kaohsiung 807, Taiwan; 3Center for Cancer Research, Kaohsiung Medical University, Kaohsiung 807, Taiwan; 4Department of Surgery, Kaohsiung Medical University Hospital, Kaohsiung 807, Taiwan; 5Department of Cosmetic Science and Institute of Cosmetic Science, Chia Nan University of Pharmacy and Science, Tainan 717, Taiwan; 6Division of Gastroenterology, Department of Internal Medicine, Kaohsiung Medical University Hospital, Center for Cancer Research, Kaohsiung Medical University, Kaohsiung 807, Taiwan; 7National Institute of Cancer Research, National Health Research Institutes, Tainan 704, Taiwan; 8Drug Development and Value Creation Research Center, Kaohsiung Medical University, Kaohsiung 807, Taiwan; 9Department of Medical Research, Kaohsiung Medical University Hospital, Kaohsiung 807, Taiwan

**Keywords:** head and neck cancer, telomerase reverse transcriptase (TERT), promoter mutations, prognosis

## Abstract

Head and neck cancers (HNCs) are among the ten leading malignancies worldwide. Despite significant progress in all therapeutic modalities, predictive biomarkers, and targeted therapies for HNCs are limited and the survival rate is unsatisfactory. The importance of telomere maintenance via telomerase reactivation in carcinogenesis has been demonstrated in recent decades. Several mechanisms could activate telomerase reverse transcriptase (TERT), the most common of which is promoter alternation. Two major hotspot TERT promoter mutations (C228T and C250T) have been reported in different malignancies such as melanoma, genitourinary cancers, CNS tumors, hepatocellular carcinoma, thyroid cancers, sarcomas, and HNCs. The frequencies of TERT promoter mutations vary widely across tumors and is quite high in HNCs (11.9–64.7%). These mutations have been reported to be more enriched in oral cavity SCCs and HPV-negative tumors. The association between TERT promoter mutations and poor survival has also been demonstrated. Till now, several therapeutic strategies targeting telomerase have been developed although only a few drugs have been used in clinical trials. Here, we briefly review and summarize our current understanding and evidence of TERT promoter mutations in HNC patients.

## 1. Introduction

Head and neck cancers (HNCs) arising in the oral cavity, oropharynx, larynx, and hypopharynx were the seventh most common cancer worldwide in 2018 [1] and approximately seven to eight hundred thousand new cases are diagnosed each year worldwide [2]. The most common type, squamous cell carcinoma, is a highly lethal group of heterogeneous neoplasms often diagnosed at an advanced stage [3]. Tobacco and alcohol consumption are the main etiological factors [4]. Betel quid chewing and infection by oncogenic human papillomavirus (HPV) types 16 and 18 have emerged as important etiological factors for a subset of HNCs in the oral cavity and oropharynx, respectively [5,6,7,8,9,10]. HPV-positive malignancies represent 5–20% of all HNCs and 40–90% of those arising from the oropharynx [11]. The prevalence of HPV-driven HNCs has been dramatically increasing in developed countries, predominantly affecting middle-aged white men, non-smokers, non-drinkers, or mild-to-moderate drinkers with a higher socioeconomic status and better performance statuses than those with HPV-unrelated SCCs [11,12]. The treatment of HNC is generally multimodal, including surgery, chemotherapy, and radiotherapy, and differs according to disease stage, anatomical location, and surgical accessibility. However, despite significant progress in all therapeutic modalities, the 5-year overall survival (OS) rate of HNC patients remains unsatisfactory [13,14,15].

In the era of biomarker-driven personalized cancer therapy, several biomarkers have been proposed as prognostic and predictive factors in different cancers, such as KIT mutations in gastrointestinal stromal tumors, EGFR mutations in lung cancer, and HER2 overexpression in breast cancer [16]. However, unlike other cancer types, there are limited predictive biomarkers and targeted therapies for HNCs [4,16,17,18]. With the development of advanced technical approaches, genome, and exome analyses have provided a comprehensive view of genetic alterations in HNC and uncovered potential new therapeutic opportunities [17,19,20,21,22,23,24]. In addition to commonly mutated genes, such as *TP53, CDKN2A, CCND1, PIK3CA,* and *NOTCH1*, telomerase reverse transcriptase (*TERT*) promoter mutations have been detected in a significant proportion of HNC patients [3,4,12,13,15,25,26,27,28,29]. *TERT* is located on chromosome 5p15.33 in humans and is an integral and essential part of the telomerase holoenzyme, which plays a key role in cancer formation. Mostly, telomerase activity was increased by upregulation of *TERT* expression via several genetic and epigenetic alterations, and *TERT* promoter mutations are known as the most important [30]. However, the incidence of *TERT* promoter mutations varies in the head and neck subsites, and the association between *TERT* promoter mutations and outcomes is unclear. Therefore, in this review, we summarize our current understanding and evidence of *TERT* promoter mutations in HNC patients.

## 2. Telomeres and Telomerase in Normal Cells

Telomeres are the physical ends of eukaryotic linear chromosomes [31,32]. In human cells, telomeres are composed of TTAGGG nucleotide repeats with a 3′ single-stranded overhang, and the variation ranges from 3 to 20 kilobase pairs [12,33]. They are bound by a six-member protein complex known as shelterin. Telomeres cover the coding DNA at the end to avoid loss of genetic information in linear DNA, act as a cap to prevent degradation by a nucleolytic attack, and prevent aberrant activation of a DNA damage response (DDR), which could lead to inappropriate processing of telomeres and as sites for double-strand break repair [31,34,35].

Telomerase, a specialized reverse transcriptase, is a large multi-subunit ribonucleoprotein complex that synthesizes telomeric DNA sequences and provides a molecular basis for unlimited proliferative potential [36]. Telomerase comprises two major components: the telomeric RNA component (also known as the telomerase RNA component: TERC or TR) and the telomerase reverse transcriptase (TERT), which is encoded by the TERT gene. The TERT is located in the human chromosome band 5p15.33, and the TERC is located at 3q26.3 [37]. The TERC serves as the template for telomere hexamer repeat additions onto the DNA, and the TERT is responsible for the reverse transcribing hexamer repeats onto the chromosomal ends [33,38,39,40]. TERT expression is silenced during development, unlike the TERC and other constitutively expressed telomerase components [35].

Telomerase is a key telomere length maintenance mechanism and is present in germline, hematopoietic, stem, and other rapidly renewing cells [41]. However, in most normal somatic cells, telomerase activity is extremely low or absent. Therefore, loss of telomeric repeats occurs at each round of DNA replication, after which the telomeres are reduced to a critical length [42]. Critical telomere attrition elicits a DDR that mediates cell cycle arrest and leads to replicative senescence or apoptosis via the p53 or Rb tumor suppressor pathways [35,43]. Telomere attrition acts as a barrier to replicative immortality, also called a “mitotic clock” that limits the cell cycle number and further triggers cellular senescence [33,35]. Although rare, in the absence of telomerase, some cells employ another DNA recombination mechanism, termed alternative lengthening of telomeres (ALT), which reverses telomere attrition to bypass senescence [44].

## 3. Telomere and Telomerase in Cancer Cells

Telomere length and telomerase activity are crucial for cellular immortalization, tumorigenesis, and cancer progression. Telomere maintenance via telomerase reactivation is a nearly universal hallmark of cancer cells [35,45,46]. The vast majority of cancers overcome replicative senescence by upregulating TERT expression and telomerase activity [35]. Telomerase activity is upregulated in 80–90% of malignancies, enabling unlimited replication of cancer cells, similar to embryonic and stem cells [13,15,47]. For the remaining 10–15% of cancers, upregulation of telomerase activity is achieved through the ALT pathway [48].

There are several ways to upregulate telomerase activity and activate the normally silent human TERT (hTERT) gene. The mechanisms of hTERT activation include chromosomal rearrangements (i.e., duplications, amplifications, insertions, interchromosomal changes, inverted orientations, or deletions), TERT promoter somatic mutations, epigenetic modifications (i.e., DNA methylation, or post-transcriptional regulation by microRNAs), transcriptional activators or repressors, TERT gene polymorphism and alternative splicing (i.e., pre-mRNA alternative splicing of the TERT gene) [30,49,50,51]. In a pan-cancer genomics study, Barthel et al. detected TERT expression in 73% of the 6835 total tumor samples, which were associated with TERT point mutations, rearrangements, DNA amplifications, and transcript fusions. Among the TERT-expressing samples, there were 31% TERT promoter mutations, 3% TERT amplifications, 3% TERT structural variants, 5% TERT promoter structural variants, and 53% TERT promoter methylation [51]. Some of these mechanisms may interact with each other and have a synergistic effect on TERT expression [30].

Besides the canonical role of telomerase in telomere maintenance, there are also some non-canonical functions (telomere length-independent mechanisms) in tumorigenesis, such as the regulation of metabolic mechanisms, epigenetic regulation, and modulation of chromatin, oxidative stress protection, RNA silencing, signal transduction pathways (Wnt and c-MYC signaling pathways), enhanced mitochondrial function, cell adhesion, and migration [30,52,53,54,55,56,57,58].

## 4. Telomerase Reverse Transcriptase (TERT) Promoter Mutations

Among the several mechanisms of hTERT activation, TERT somatic promoter mutations are the most common non-coding driver mutations in cancer [30,59] and occurred at a high frequency in over 50 cancer types [60]. They have been reported in two major hotspots (mainly C ˃ T transitions), which are located at −124 and −146 base pairs upstream of the transcriptional start site on chromosome 5 and are designated as C228T and C250T, respectively [13,25,32,61,62]. A less frequent hTERT promoter mutation −57 base pairs upstream of the transcriptional start site with an A > C transition (at position 1,295,161 on chromosome 5) has been found to be a disease-segregating germline mutation in a melanoma-prone family [63]. Other less frequent, yet recurrent, mutations on chromosome 5 have also been discovered in cancers at the following positions: 1,295,228 C > A, 1,295,248–1,295,243 CC > TT, and 1,295,161 A > C [64].

TERT somatic promoter mutations are predominantly heterozygous and lead to the generation of an 11 bp sequence, CCCGGAAGGGG, which is similar to the E26 transformation-specific (ETS) factor binding motif [60,65]. Then, ETS binding factors, such as GA-binding protein (GABP), are recruited. This recruitment resulted in direct transcriptional activation of hTERT expression and promoted an epigenetic shift from a repressed to active chromatin conformation [35,65,66,67,68]. These promoter mutations were proven to be associated with higher levels of TERT mRNA, TERT protein, telomerase enzymatic activity, and telomere length in a study of 23 human urothelial cancer cell lines [69].

Two TERT promoter hotspot mutations, C228T and C250T are the most common; however, their frequencies vary widely across tumors from different sites (Table 1). These mutations occur most frequently in cancers with low rates of self-renewal [25] and are rare in pediatric and young adult cancers [3,70]. The highest frequencies of TERT promoter mutations have been reported in melanoma, bladder cancer, urothelial carcinoma, CNS tumors, hepatocellular carcinoma, thyroid cancer, basal cell carcinoma, and cutaneous squamous cell carcinoma. Due to the variety of sarcoma subtypes, the prevalence of TERT promoter mutations varies widely, and the highest TERT promoter mutation rate is reported in myxoid liposarcoma (79.1%) [25].

## 5. TERT Promoter Mutations in Head and Neck Squamous Cell Carcinoma

### 5.1. The Frequency of TERT Promoter Mutations

For HNCs, the frequency of TERT promoter mutations varied significantly among previous studies. These differences could be explained by the tumor subsite, sample size, methodological sensitivity, risk factors, and population ethnicity (Table 2).

Killela et al. surveyed 70 oral cavity cancers and identified TERT promoter mutations in 12 of the tumors (17.1%) [25]. Schwaederle et al. analyzed 423 cases of TERT promoter alterations using next-generation sequencing (NGS). Only 28 patients (6.6%) had HNCs. The incidence of TERT promoter alternations was 14.4% (61 of 423) in the overall population and 28.6% (8 of 28) in HNCs [32]. Cheng et al. collected 84 cases of SCC from different sites, including 12 HNC and C228T mutations, which were detected in 16.67% (2 of 12) [93]. Barczak et al. analyzed 61 HNC patients to determine the prevalence of the hTERT promoter C250T mutation. High-resolution melting mutation analysis was used to identify the C250T hTERT promoter mutation, followed by sequencing verification in 10% of the samples. The prevalence of the hTERT promoter C250T mutation was 36% [15]. Yu et al. identified TERT promoter mutations in 117 patients with SCC of the oral cavity (N = 74), larynx (N = 24), hypopharynx (N = 5), and HPV-negative SCC of the oropharynx (N = 14) using NGS. Overall, 63 patients (53.8%) had TERT promoter alterations, and the most common mutations were C228T and C250T [29]. Morris et al. collected 53 patients and 20 oral cavities, 18 oropharynges, 7 larynges, and 2 hypopharynges. Overall, the frequency of the TERT alteration was 32.1% (17 of 53), yet it was much higher in HPV-negative tumors (53% vs. 4.3%). Remarkably, 91% (10 of 11) of the HPV-negative tongue SCCs possessed TERT mutations [97].

In Italy, Boscolo-Rizzo et al. analyzed cancer tissue and adjacent mucosa specimens from 101 patients with HNCs and evaluated the prevalence of the TERT promoter mutations by Sanger sequencing. The tumor subsites in the HNCs included the oral cavity (N = 27), oropharynx (N = 23), hypopharynx (N = 15), and larynx (N = 36). The TERT promoter harbored mutations in 12 tumors (11.9%), with C228T and C250T, which accounted for 83.3% and 16.7%, respectively. They also evaluated the TERT mRNA level and found no significant difference between the TERT mRNA level and the mutational status of the TERT promoter [3]. Annunziata et al. analyzed tumor biopsies from 15 oral SCCs and nine oropharyngeal SCCs. The frequency of TERT promoter mutations was 60% (9 of 15) in oral SCCs and was absent in oropharyngeal SCCs. There were five hotspot mutations (three C228T and two C250T) and four other mutations. They also investigated the TERT mRNA levels and identified that the TERT mRNA levels were comparable to those detected in peri-tumor tissues. However, these data were from six oropharyngeal SCCs and illustrated that they all lacked mutations in the TERT promoter [96].

In Turkey, Yilmaz et al. collected a total of 189 patients with HNCs, including 102 oral cavities, 22 oropharynges, 6 hypopharynges, and 59 larynges. The TERT gene expression was examined by polymerase chain reaction (PCR)-based direct sequencing. TERT promoter mutations were detected in 43.9% (83 of 189) of the cases. Three TERT promoter region mutations were detected: C228T (56 of 83; 67.5%), C250T (22 of 83; 26.5%), and C228A (5 of 83; 6%). The frequency of the C228T mutation was almost twice that of the C250T and C228A mutations [4]. In Brazil, Arantes et al. collected 88 HNC patients and analyzed the TERT promoter mutations C228T and C250T using pyrosequencing. The overall prevalence of the TERT hotspot mutations is 27.3% (6.8% at locus C228T and 20.5% at C250T) [13]. In India, Vinothkumar et al. analyzed 181 primary tumors of the uterine cervix and oral cavity using PCR amplification and sequencing. A high frequency of TERT hotspot mutations was observed in both cervical (30 of 140, 21.4%) and oral (13 of 41, 31.7%) SCCs. Among the oral cancer samples, the TERT promoter hotspot mutations were frequent, while the C228T mutation (69.2%) was twice as frequent as the C250T (30.8%) [26].

In Taiwan, Chang et al. included 201 oral cavity SCC tumors and adjacent normal tissues to detect two TERT promoter mutations (C228T and C250T) using Sanger sequencing. Overall, the TERT hotspot promoter mutations occurred at a high frequency (64.7%) in patients with oral cavity SCCs. There were 52.5% (104 of 201) and 12.9% (26 of 201) oral cavity SCC tumor tissues containing that contained the C228T and C250T mutations, respectively [28].

In China, Qu et al. obtained 235 laryngeal cancer tissues using a pyrosequencing assay to detect the TERT promoter mutations C228T and C250T. The TERT promoter hotspot mutations were present in 27% (64 of 235) of the samples. The TERT C250T mutations were more common (56 of 235) than the C228T mutations (8 of 235) [27]. In Figure 1, we summarized the reported frequencies of the TERT promoter mutations in HNCs from the various studies mentioned above.

### 5.2. TERT Promoter Mutations in Different Anatomic Distribution

HNC is a heterogeneous group of tumors involving distinct anatomical sites and subsites with varying etiological factors. Yu et al. showed that TERT promoter mutations were more abundant in oral cavity SCCs than in laryngopharyngeal cancers (81.1% vs. 7.0%) [29]. Boscolo-Rizzo et al. demonstrated that the prevalence of TERT hotspot promoter mutations is significantly higher in oral cavity SCCs (37%) [3]. Annunziata et al. also showed that TERT promoter mutations were predominant in oral SCCs (60%), yet absent in oropharyngeal SCCs [96]. Arantes et al. reported that 92% of the mutation cases were located in the oral cavity [13]. Finally, Yilmaz et al. showed that the frequency of the TERT promoter mutations in oral SCCs (75.5%) was significantly higher than in the other locations [4]. The anatomic distribution of cases is strongly associated with TERT promoter mutations, and the highest frequency is in oral cavity cancers.

As for the subsites in oral cavity SCCs, Arantes et al. noticed that 92% of the mutated cases were mainly in the tongue [13]. Killela et al. also revealed that 11 out of the 12 cancers with TERT promoter mutations were in the oral tongue, although only 23 of the 70 oral cavity cancers originated in the oral tongue [25]. However, Yilmaz et al. demonstrated that the highest rate was related to the buccal location and the lowest to the floor of the mouth (82.35% and 61.53%, respectively), although the difference was not statistically significant [4].

### 5.3. TERT Promoter Mutation and Human Papillomavirus Status

An association between HPV infection and oropharyngeal SCC has been proven. It was also clear that the molecular landscape and clinical pattern were different between HPV-positive and HPV-negative oropharyngeal cancers [10]. Only two studies have investigated the association between HPV status and TERT promoter mutations.

In a cohort of 53 patients with advanced HNCs, performed by Morris et al., a very high TERT alternation rate (53%, 16 of 30) was present in 30 HPV-negative tumors, however, there was only one TERT alternation (4.3%), which was a TERT amplification rather than a hotspot mutation, in 23 HPV-positive tumors. HPV-negative tongue SCCs showed the highest TERT mutation rate (91%). This demonstrated that TERT mutations and HPV infection may represent parallel mechanisms of telomerase activation in HNCs [97]. In another cohort study conducted by Annunziata et al., among the 9 patients with TERT promoter mutations in 15 oral SCC patients, 7 were HPV-negative and 2 were HPV-positive (*p* = 0.486). The frequency of TERT mutations was also independent of HPV tumor status in oral cancer [96].

### 5.4. TERT Promoter Mutation and Tobacco, Alcohol, and Betel Quid

Aside from HPV infection, tobacco smoking, alcohol consumption, and betel quid chewing are the other three main etiological factors of HNC [4,5]. Until now, the relationship between the TERT promoter mutations and these three factors remains inconclusive.

In a Brazilian cohort of 88 patients with HNC conducted by Arantes et al., the frequency of the C250T mutation appeared to be higher in alcohol consumers. Of the patients harboring the TERT promoter mutation C250T, 94.4% were alcohol consumers, and 66.7% of the patients harboring the TERT promoter mutation C228T did not consume alcohol [13]. In a Chinese cohort of 235 laryngeal cancer cases reported by Qu et al., hotspot mutations were not significantly correlated with any clinicopathological variables. However, TERT promoter mutations, particularly the C250T mutation, were more frequent in smoking patients (47 of 130) than in non-smoking patients (9 of 49), although no statistical significance was noted [27]. In a cohort of 201 patients with oral cavity SCC performed by Chang et al. in Taiwan, the C228T mutation was significantly associated with betel nut chewing [28]. In contrast, in a Turkish cohort of 189 HNC patients performed by Yilmaz et al., TERT promoter region mutations in HNC were inversely related to smoking and alcohol consumption [4].

### 5.5. TERT Promoter Mutation and Other Factors

Schwaederle et al. demonstrated that TERT promoter alterations are more frequent in men. They were also associated with brain cancers, skin/melanoma, head, and neck tumors, and increased median numbers of alterations in the univariate analysis. However, this association in head and neck tumors was not found in further multivariate analyses [32]. Yilmaz et al. reported that TERT promoter region mutations in HNCs are associated with younger age and female genders in a cohort from Turkey [4]. Barczak et al. demonstrated a significant association between the frequency of the homozygous C250T mutation and tumor grade (T1 = 27%, T2 = 36%, T3 = 35%, T4 = 46%, *p* ≤ 0.0001) [15]. However, in a cohort of 41 patients with oral SCCs, performed by Vinothkumar et al. in India, no significant correlation was observed between any of the genotypes and the clinicopathological characteristics [26].

### 5.6. TERT Promoter Mutation and Survival

TERT promoter mutations in various reports of different cancers have been associated with aggressive characteristics, poor outcomes, and shorter survival [98,99,100,101]. In HNC, Qu et al. showed that TERT promoter mutations significantly affected the overall survival of laryngeal cancer patients, particularly those with the C250T mutation. TERT promoter mutations were significant predictors of poor prognosis in patients with laryngeal cancer, as an independent variable, with respect to age, tumor localization, TNM stage, tumor invasion, lymph node metastasis, and smoking history [27]. Schwaederle et al. also demonstrated a significantly shorter overall survival in patients harboring the TERT promoter alterations in the overall population in a univariate analysis. Subanalyses of the three tumor types with the highest prevalence of TERT alterations consistently showed a trend toward shorter survival for patients with altered TERT promoters in brain tumors, head, and neck cancers, and melanoma/skin tumors [32]. Arantes et al. demonstrated no statistically significant association between the presence of hotspot mutations (C228T and C250T) and survival. However, the presence of the C228T mutation impacted patient outcomes, with a significant decrease in 5-year disease-free survival (20.0 vs. 63.0%) and 5-year overall survival (16.7 vs. 45.1%) [13].

Similar results were reported by Yu et al. [29]. They reported that the TERT promoter mutations were associated with locoregional failure (LRF) in the overall cohort and in oral cavity SCCs. This increased risk for LRF is independent of the oral cavity primary site, TP53 mutation status, extracapsular extension, and positive surgical margins suggesting that the TERT promoter mutations are an independent biomarker of LRF rather than a surrogate for OSCCs, or other known prognostic markers. The cumulative incidence of LRF was similar between the two types of TERT promoter mutations (C250T and C228A/T groups), and both were associated with a higher cumulative incidence of LRF compared to wildtype tumors. Overall, they demonstrated that TERT promoter mutations were associated with an increased risk of LRF, although not with distant failure or overall survival [29].

In contrast, Yilmaz et al. did not find a significant association between the presence of TERT mutations and OS, despite patients with HNCs harboring TERT mutations exhibiting a slightly shorter median OS [4]. Boscolo-Rizzo et al. showed no significant association between the TERT promoter status and overall survival, although the TERT mRNA level had an impact on clinical outcomes [3]. Chang et al. also reported that there was no significant difference in overall survival, disease-specific survival, and disease-free survival between TERT promoter mutations and the wildtype [28].

## 6. Anti-Telomerase Therapeutics

The unique feature of overexpression in most cancer cells, although absent or with low expression in somatic cells, makes telomerase and other telomere components a target for the development of therapeutics [30]. Several therapeutic strategies have been proposed to target telomerase, and some have already been evaluated in clinical trials against various cancer types [30,35,44,102]. However, the development of successful clinical therapies is hampered by significant challenges [35].

### 6.1. Direct Telomerase Inhibition

Direct telomerase inhibition by small molecules or oligonucleotides that directly bind to the TERT or TERC template region suppresses telomere extension.

The first-in-class modified oligonucleotide, GRN163L (Imetelstat), was developed in 2003 [103]. Imetelstat is a lipidated 13-mer thiophosphoramidate oligonucleotide complementary to the TERC template region, which competitively inhibits telomerase activity and suppresses cancer cell viability [103]. After showing activity and efficacy against multiple cancer cell lines and in mouse xenograft models, Imetelstat has moved to early clinical trials against solid tumor malignancies (such as breast cancer, non-small-cell lung cancer, brain tumor, and melanoma) and hematologic diseases (such as multiple myeloma, myelodysplastic syndrome, and myeloproliferative neoplasms) [35,44,102]. Although Imetelstat did not meet its efficacy endpoints in trials on non-small cell lung cancer and breast cancers [104,105], it showed robust response rates in patients with lower-risk myelodysplastic syndromes, myelofibrosis, and essential thrombocythemia [106,107,108,109], and further late-stage clinical trials are underway [110].

BIBR1532, 2-[[(E)-3-naphthalen-2-ylbut-2-enoyl]amino]benzoic acid, inhibits telomerase by non-competitively binding to the TERT active site [111]. BIBR1532 has generated promising preclinical results [112,113,114,115,116,117,118]. For example, it enhances the radiosensitivity of non-small cell lung cancer by increasing telomere dysfunction and ATM/CHK1 inhibition [117]. However, this has not yet progressed to clinical testing. Some natural compounds have also been reported to act as telomerase inhibitors, such as allicin (from garlic), curcumin (from turmeric), silibinin (from thistle), and epigallocatechin gallate (EGCG, from tea), and the EGCG’s derivative, MST-312 [30,35,119].

### 6.2. G-Quadruplex Stabilizers

G-quadruplexes are tetrad planar structures formed in guanine-rich DNA or RNA sequences, including telomeres [120]. Compounds that would stabilize telomeric G-quadruplex secondary structures can disrupt telomere extension via telomerase, triggering a DNA damage response and cell death. Several G-quadruplex stabilizers, including telomestatin, BRACO-19, RHPS4, TMPyP4, CX-3543 (Quarfloxin), CX-5461 (Pidnarulex) and AS1411, have been tested in preclinical studies and some already progressed to clinical trials [30,102,121,122,123,124,125,126,127,128,129,130].

### 6.3. Nucleoside Analogues

Nucleoside analogs mimic the presence of uncapped telomeres and induce DNA damage response, apoptosis, and autophagy [102]. To date, several nucleotide analogs, including T-oligo, 6-thio-2′-deoxyguanosine (6-thio-dG), and 5-fluoro-2′-deoxyuridine (5-FdU) triphosphate, are under investigation, although have not yet advanced to clinical trials [131,132,133,134].

### 6.4. Telomerase-Based Cancer Vaccines

Telomerase-based therapeutic cancer vaccines aim to induce T cells that target a tumor antigen, leading to improved antitumor immune responses and cancer cell death [135]. TERT is an appropriate tumor-associated antigen. To date, telomerase vaccinations, including peptide vaccines (such as GV1001, GX301, UV1, and Vx-001), dendritic cell-based vaccines (such as GRNVAC1), and DNA vaccines (such as INVAC-1) have been evaluated in many clinical trials spanning almost two decades [35,44,102,135,136]. Clinical studies on hTERT have been applied to both solid tumors and hematologic malignancies, and some of them have already moved to the later stages of trials [136]. However, the efficacy of the TERT vaccines is insufficient [137,138]. Furthermore, therapeutic TERT-based vaccines can mediate specific T cell responses in a high proportion of cancer patients [35]. A more robust antitumor activity was observed when combining immune checkpoint blockade with TERT-based vaccines in preclinical research, proving the synergistic effect between these two drugs [139].

### 6.5. TERT or TERC Promoter-Driven Therapy

Owing to the hallmark role of TERT promoter mutation-induced TERT expression in tumorigenesis, correction of this mutation and reduction of TERT expression has become a therapeutic method, by using recently developed gene editing techniques, including oncolytic virus and suicide gene therapy [102]. Telomelysin (OBP-301), a telomerase-specific replication-component adenovirus with an hTERT promoter element, has shown strong antitumor effects in various human cancer cells, including HNCs [140,141]. Phase I trials for solid tumors [142] and advanced hepatocellular carcinoma [143] have already been completed. Further phase 2 trials on gastric/gastroesophageal junction cancers (NCT03921021), head and neck cancers (NCT04685499), and esophageal cancers (NCT03213054) are currently ongoing.

### 6.6. Other Therapeutics Strategies

In addition to the strategies mentioned above, there are other anti-telomerase therapies. For example, telomerase interference by altered TERC templates is introduced by lentiviral infection [144], CRISPR genome editing targeting TERT gene expression [145], inhibition of oncogenic signaling MAPK pathways that impinge on TERT transcription [35], epigenetic mechanisms using histone deacetylase [30,146], and Tankyrase inhibitors for telomere length regulation [102,147]. Furthermore, strategies relying on telomere attrition in the setting of adjuvant or maintenance therapies rather than frontline therapy are another consideration [35].

## 7. Conclusions

Telomeres shorten with each cell division and result in cellular senescence. Telomere maintenance via telomerase reactivation plays a critical role in tumorigenesis (Figure 2). Several mechanisms could increase telomerase activity and TERT promoter, C228T and C250T, mutations are the most well-known alternations, which have already been reported in several malignancies, including HNCs. The frequency of the TERT promoter mutations in HNCs is quite high, ranging from 11.9% to 64.7%, and is more enriched in oral cavity SCCs than in other subsites. In addition, several reports have demonstrated an association between TERT promoter mutations and poor survival. Over the past 10 to 20 years, several anti-telomerase therapeutic strategies have been developed. However, only a few drugs have been used in clinical trials, and the results are only passable. Studies on immunotherapies targeting telomerase, such as cancer vaccines and oncolytic viruses, are very promising, however, trials are still ongoing. In addition, the potential synergy between TERT vaccines and immune checkpoint blockade may be another way to maximize anti-telomerase therapy.

## Figures and Tables

**Figure 1 biomedicines-11-00691-f001:**
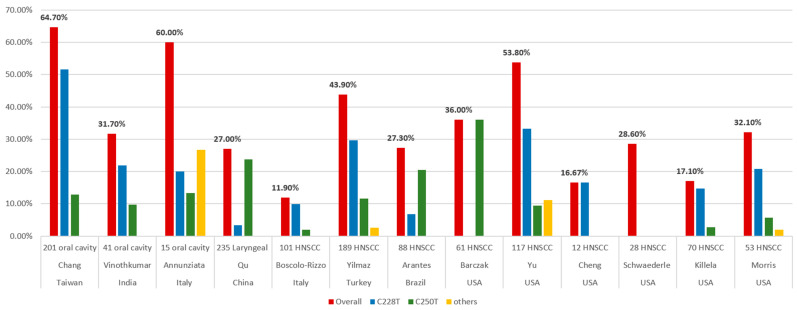
Frequencies of TERT promoter mutations in head and neck cancers from different studies.

**Figure 2 biomedicines-11-00691-f002:**
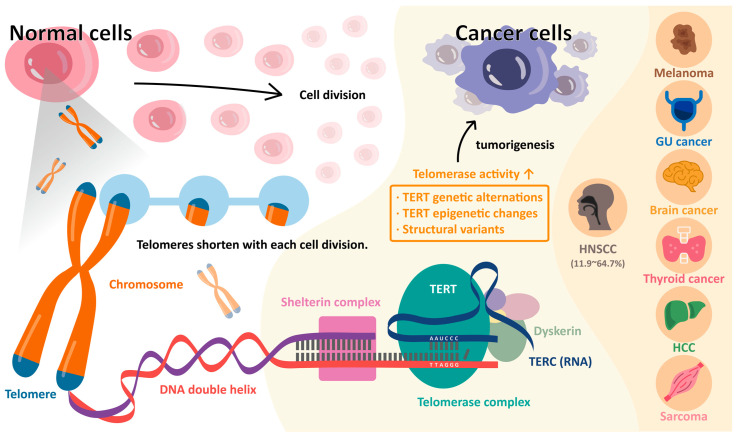
Telomere and telomerase play important roles in cellular biology and tumorigenesis. Telomeres are specialized structures that are located at the ends of chromosomes. They are composed of DNA repeating sequences (TTAGGG). Shelterin complexes are specific proteins known to protect chromosomes and regulate telomere length. Telomerase is a reverse transcriptase that synthesizes telomeric DNA sequences to maintain telomere length. Telomerase comprises two major components: the telomeric RNA component (TERC) and the telomerase reverse transcriptase (TERT). Other proteins, such as dyskerin, are also found in a complex with TERC. Telomeres shorten with each round of cell division and this mechanism limits the proliferation of cells to a finite number of cell divisions. Unlike normal cells, cancer cells are characterized by high telomerase activity, which could be achieved via mechanisms including TERT genetic alterations, TERT epigenetic change, or structural variants. TERT promoter mutations are the most common alternation and have been reported in several malignancies such as melanoma, genitourinary cancers, CNS tumors, hepatocellular carcinoma, thyroid cancers, sarcomas, and HNCs. In HNCs, the frequency of TERT promoter mutations is high (11.9–64.7% based on different studies).

**Table 1 biomedicines-11-00691-t001:** Frequency spectrum of hTERT promoter mutations across different cancer types.

Cancer Type	Mutation Frequency (%)	Reference
Malignant melanoma	17.0–85.0	[61,63,71,72]
Genitourinary cancers		
Bladder cancer	59.0–85.0	[25,61,73,74,75,76]
Urothelial carcinomas	29.5–64.5	[77,78]
Kidney cancers	0	[61]
Prostate Cancer	0	[79]
CNS tumors		
Glioblastoma	54.0–84.0	[61,70,73,78,80]
Other gliomas (ependymoma, astrocytoma, mixed glioma, oligodendroglioma)	2.7–78.0	[25,64,70,78]
Medulloblastoma	33.3–65.0	[70,78]
Hepatocellular carcinoma	31.4–59.0	[25,78,81,82,83,84]
Thyroid cancer (papillary, follicular, poorly differentiated, and anaplastic carcinomas)	3.4–46.3	[61,85,86,87]
Gastrointestinal stromal tumor	0–3.8	[61,88]
Malignant pleural mesothelioma	11.3	[89]
Atypical fibroxanthomas	93.0	[90]
Sarcomas (chondrosarcoma, fibrosarcoma, myxofibrosarcoma, myxoid liposarcoma, osteosarcoma, pleomorphic dermal sarcomas)	4.3–79.1	[25,90,91]
Basal cell carcinoma of the skin	73.8	[92]
Squamous cell carcinoma of the skin	20.0–74.0	[25,92,93]
Squamous cell carcinoma of esophageal	1.6	[94]
Squamous cell carcinoma of penile	48.6	[95]
Squamous cell carcinoma of the head and neck	11.9–64.7	[3,4,13,15,25,26,27,28,29,32,93,96,97]
Squamous cell carcinoma of the cervix	0–21.4	[25,26,93,96]
Breast cancer, colorectal cancer, ovarian cancer, esophageal adenocarcinoma, acute myeloid leukemia, chronic lymphoid leukemia, pancreatic cancer, and testicular carcinoma	0–5.0	[61,78]

**Table 2 biomedicines-11-00691-t002:** Summary of studies evaluating the association between head and neck cancers with TERT promoter mutations.

Author, Country (Year)	Case Numbers	Cancer Sites	Prevalence of TERT Promoter Mutations	Special Findings	The Association with Survival
Killela, USA (2013) [25]	70	31 Oral cavity 23 Oropharynx 4 Supraglottic 12 Others	Total: 17.1% (12/70) C228T: 14.8% C250T: 2.8%	Highest frequency in tongues (47.8%, 11/23)	N/A
Schwaederle, USA (2018) [32]	28	28 HNC	Total: 28.6% (8/28)	N/A	Trend toward shorter survival
Cheng, USA (2015) [93]	12	12 HNSCC	Total: 16.67% (2/12) C228T: 16.67% C250T: 0%	No significant correlation was observed.	N/A
Barczak, USA (2017) [15]	61	25 Mouth 25 Voice box 5 Nose/sinuses 6 Throat	C250T homozygous T/T allele: 36% heterozygous C/T allele: 26%	Homozygous T/T mutation is associated with the grade of the tumor.	N/A
Yu, USA (2021) [29]	117	74 Oral cavity 24 Larynx 5 Hypopharynx 14 HPV (-) oropharynx	Total: 53.8% (63/117) C228T: 33.3% C250T: 9.4% C250T, C254T: 6% C228A: 4.3% CC434TT: 0.9%	Highest frequency in the oral cavity (81.1%, 60/74)	Increased risk of locoregional failure, but not distant failure or OS.
Morris, USA (2017) [97]	53	20 Oral cavity 18 Oropharynx 7 Larynx 2 Hypopharynx 6 Others (4 sinonasal cavity)	Total: 32.1% (17/53) C228T: 20.8% C250T: 5.7% C228A: 1.9%	TERT mutation and HPV infection may represent parallel mechanisms.	N/A
Boscolo-Rizzo, Italy (2020) [3]	101	27 Oral cavity 23 Oropharynx 15 Hypopharynx 36 Larynx	Total: 11.9% (12/101) C228T: 9.9% C250T: 2%	Highest frequency in the oral cavity (37%) TERT levels did not significantly differ according to the mutational status of TERT promoter.	No significant association between TERT promoter status and OS. Higher TERT levels, worse OS (43.6% vs. 60.1%)
Annunziata, Italy (2018) [96]	24	15 Oral cavity 9 Oropharynx	Total: 37.5% (9/24) C228T: 8.3% C250T: 12.5% Other: 16.7%	No mutation in oropharynx cancer. Mutations were independent of HPV status.	N/A
Yilmaz, Turkey (2020) [4]	189	102 Oral cavity 22 Oropharynx 6 Hypopharynx 59 Larynx	Total: 43.9% (83/189) C228T: 29.6% C250T: 11.6% C228A: 2.6%	Highest frequency in the oral cavity (75.5%, 77/102). TERT mutations are associated with younger age, female gender, and an inverse relationship to smoking and alcohol consumption.	No difference
Arantes, Brazil (2020) [13]	88	69 Oral cavity 11 Larynx 8 Pharynx	Total: 27.3% (24/88) C228T: 6.8% C250T: 20.5%	94.4% C250T were alcohol consumers. 66.7% C228T were not alcohol consumers	Decreased 5-year DFS and OS in C228T
Vinothkumar, India (2016) [26]	41	41 Oral cavity	Total: 31.7% (13/41) C228T: 21.9% C250T: 9.7%	No significant correlation was observed.	N/A
Chang, Taiwan (2017) [28]	201	201 Oral cavity	Total: 64.7% (130/201) C228T: 51.7% C250T: 12.9%	C228T mutation was associated with betel nut chewing.	No difference
Qu, China (2014) [27]	235	235 Laryngeal	Total: 27% (64/235) C250T: 23.8% C228T: 3.4%	Not significantly correlate with any clinicopathological variables	Poor survival, especially C250T mutation

## Data Availability

The data that support the findings of this study are available upon reasonable request (e.g., for research purposes) from the authors.

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
