# Peer review of "Deciphering the Functions of Telomerase Reverse Transcriptase in Head and Neck Cancer"

_biomedicines, 2023, doi:10.3390/biomedicines11030691_

Round 1

Reviewer 1 Report

This review lacks a proper introduction/description of the evidence for the re-expression of TERT by these mutations in the promoter in the TERT gene in various cancers.  

Do these mutations in the TERT promoter alsways result in the re-expression of TERT?  What degrees of the re-expression were observed associated with these mutations? Are these levels of the re-expression biologically relevant?

It is also better to include the information on the expression status of TERT in the studies listed in Table2 if it was investigated.  

Please cite these publications properly.

Bell RJA, Rube HT, Kreig A, Mancini A, Fouse SD, Nagarajan RP, et al. Cancer. The transcription factor GABP selectively binds and activates the mutant TERT promoter in cancer. Science. 2015;348:1036–9.

Stern JL, Theodorescu D, Vogelstein B, Papadopoulos N, Cech TR. Mutation of the TERT promoter, switch to active chromatin, and monoallelic TERT expression in multiple cancers. Genes Dev. 2015;29:2219–24.

Reviewer 2 Report

In this manuscript authors study how the telomerase reverse transcriptase  ummarize their understanding and evidence of TERT promoter mutations in head and neck cancer patients, a scientifically sound issue nowadays.

This article is clear, comprehensive and well-structure. Figures and tables are easy to understand and interpret, and show the data properly.

Statements and conclusions are consistent with the evidence and arguments presented and they have relevance to the field.

References are current (only 12/142 from more than ten years ago), relevant and there is not an excessive number of self-citations.

In general, I consider this article has interest to the scientific community.

Round 2

Reviewer 1 Report

The authors responded to my comments satisfactory.